# Correlating global trends in COVID-19 cases with online symptom checker self-assessments

**Marc Zobel**[1]*, **Bernhard Knapp**[1,2], **Jama Nateqi**[3,4], **Alistair Martin**[1]

**1** Data Science Department, Symptoma, Vienna, Austria, **2** Faculty Computer Science, University of Applied Sciences Technikum, Vienna, Austria, **3** Medical Department, Symptoma, Attersee, Austria, **4** Department of Internal Medicine, Paracelsus Medical University, Salzburg, Austria

\* science@symptoma.com

## Abstract

### Background

Online symptom checkers are digital health solutions that provide a differential diagnosis based on a user's symptoms. During the coronavirus disease 2019 (COVID-19) pandemic, symptom checkers have become increasingly important due to physical distance constraints and reduced access to in-person medical consultations. Furthermore, various symptom checkers specialised in the assessment of COVID-19 infection have been produced.

### Objectives

Assess the correlation between COVID-19 risk assessments from an online symptom checker and current trends in COVID-19 infections. Analyse whether those correlations are reflective of various country-wise quality of life measures. Lastly, determine whether the trends found in symptom checker assessments predict or lag relative to those of the COVID-19 infections.

### Materials and methods

In this study, we compile the outcomes of COVID-19 risk assessments provided by the symptom checker Symptoma (www.symptoma.com) in 18 countries with suitably large user bases. We analyse this dataset's spatial and temporal features compared to the number of newly confirmed COVID-19 cases published by the respective countries.

### Results

We find an average correlation of 0.342 between the number of Symptoma users assessed to have a high risk of a COVID-19 infection and the official COVID-19 infection numbers. Further, we show a significant relationship between that correlation and the self-reported health of a country. Lastly, we find that the symptom checker is, on average, ahead (median +3 days) of the official infection numbers for most countries.

**Data Availability Statement:** All data required to reproduce the analyses presented in this manuscript can be found at https://github.com/symptoma/global_trends_symp_c19. Included are the scaled user count per country per day over the

period analysed alongside all external data sourced by the authors. The untransformed user counts and the mean number of unique users per country are available upon request via science@symptoma.com.

**Funding:** This study has received funding from the European Union's Horizon 2020 research and innovation programme under grant agreement No 830017 and by the Austrian Research Promotion Agency under grant agreement No 880939 (supported by the Federal Ministries Republic of Austria for Digital and Economic Affairs and Climate Action, Environment, Energy, 252 Mobility, Innovation and Technology).

**Competing interests:** All authors are employees of Symptoma GmbH. JN holds shares of Symptoma.

## Conclusion

We show that online symptom checkers can capture the national-level trends in coronavirus infections. As such, they provide a valuable and unique information source in policymaking against pandemics, unrestricted by conventional resources.

## Introduction

Pandemics are a unique challenge for both policymakers and the general population. The former requires various up-to-date information to make informed decisions, including locations of virus outbreaks, emerging variants, and the current load on the health system [1]. The general population instead requires infrastructure to access curated information and health services, like tests, vaccination, and consultation. The global scale of the coronavirus disease 2019 (COVID-19) pandemic has made access to in-person medical consultations, diagnosis, and guidance more difficult. Health services are often overburdened, and social distancing has become a key policy in reducing the number of infections. In turn, this has increased the need for digital health solutions, whereby these services are provided without the in-person requirement of a health professional [2].

Concerning the pandemic, digital health solutions have been applied to diagnostics, symptom monitoring, contact tracing, electronic health record screening, research, and the spread of information [2, 3]. Other surveys list the applications as screening, triage, diagnosis, monitoring, and contact tracing [4]. A prominent example of contact tracing software is the mobile app TraceTogether used in Singapore [5] to track past contacts of infected individuals. Surveillance as the continuous, systematic collection, analysis, and interpretation of health-related data needed for the planning, implementation, and evaluation of public health practice [4] is the goal of Online surveillance-mapping tools like SORMAS [6] and HealthMap [7]. Monitoring systems are usually deployed in a more private context to record the condition of already infected individuals. For example, the Sense FollowUp app [8] has been used to monitor Ebola patients remotely and raise an alarm in case of increased temperature readings. One specific type of application in the context of triage is online symptom checkers, which are experiencing an increase in popularity [9]. These checkers assess the user's health situation based on entered symptoms and return a list of potential diseases. We have shown previously that symptom checkers can be applied as accurate preliminary tests to prescreen patients for COVID-19 [10]. Following that, we compared 12 online symptom checkers specialized for COVID-19, finding a broad spectrum of performances [11]. Among these, 'Symptoma' [10] showed the best overall accuracy in discriminating COVID-19 infections from non-COVID-19 control cases that exhibited flu-like symptoms [11].

In the current study, we hypothesize that the temporal and spatial distribution of user interactions with a symptom checker should reflect the global state of health. Regarding the COVID-19 pandemic, we investigate whether the lab-confirmed numbers of new COVID-19 cases correlate with the number of high COVID-19 risk assessments given by the 'Symptoma' symptom checker. Others performed similar correlation analyses in the context of temperature [12, 13], air pollution [14], and income inequality [15]. We show that our hypothesis holds; for most countries, the count of high-risk assessments reflects the trends in COVID-19 cases nationally. Further, we relate these country-wise correlations with four different socio-economic factors and find a significant connection to self-reported health from the OECD Better Life Index 2017. Lastly, we investigate the optimal lag between the official case counts and the

symptom checker assessments to maximise correlations to indicate whether the assessments could predict the pandemic progression.

## Materials and methods

### Data

**COVID-19 cases.** COVID-19 case counts were extracted from the GitHub repository of Johns Hopkins University Center for Science and Engineering (JHU CSSE) [16]. This data contains the cumulative number of positively tested persons per country daily. Based on these numbers, we calculated the total number of daily new cases as the delta of two consecutive days between the 7th of April and the 1st of November 2020 inclusive. This period contains 208 days. We refer to this dataset within the text as 'COVID-19 cases'.

**Symptom checker assessments.** Symptoma ([www.symptoma.com](www.symptoma.com)) records the anonymized results of user sessions. Regarding COVID-19, this data consists of the likelihood of infection based on symptoms, epidemiological data, and risk factors entered by the user. Symptoma also records the date and country of the user. Further details regarding this assessment can be found in our previous work [10]. From this, we extracted the total number of daily sessions that classified a user as "high-risk" with regard to COVID-19. Only countries with sufficiently consecutive data, that being no more than eight days without a single high-risk assessment, between the 7th of April and 1st of November 2020, were included in the further analysis. This constraint results in a dataset of 18 countries containing between 200 and 208 days each. Missing values were inserted as zeros, so that the time series length of each country is 208 days. The respective countries are Australia, Austria, Belgium, Canada, the Czech Republic, France, Germany, Greece, Hungary, Ireland, Italy, Netherlands, Spain, Sweden, Switzerland, Turkey, the United Kingdom, and the United States. We refer to this data within the text as 'Symptoma risk assessments'.

**Pre-processing.** Delayed reporting of new COVID-19 cases is a well-known problem [17]. To account for this problem and remove unnatural spikes (e.g., weekends), we applied a seven-day mean filter to the COVID-19 cases and the Symptoma risk assessment. Additionally, we scaled the data such that zero equals the minimum and one equals the maximum. This transformation makes the time series comparable in magnitude and protects the sensitive user numbers from public release while not affecting any of the analyses performed in this manuscript.

**Quality of LIFE measures: OECD.** We selected four quality of life (QOL) measures to reflect the featured countries' income status, health, and relationship to technology, comparing each with the correlation between Symptoma risk assessment and COVID-19. Association was measured using adjusted $R^2$ with statistical significance based on the $F$-test. Significance was corrected for multiple testing using the Bonferroni method. The measures "self-reported health" and 'life satisfaction' are sourced from the Organisation for Economic Co-operation and Development (OECD) Better Life Index 2017 dataset [18]. This dataset compares the well-being across countries based on 11 features the OECD has identified as essential.

The OECD QOL metric "self-reported health" is the percentage of the population aged 15 years or over who reported their health as 'good' or better in response to the question "How is your health in general?". The response scale is ['very bad', 'bad', 'fair', 'good', 'very good']. One should note that participants from the United States were presented with a different reporting scale which may lead to an upward bias in the reported value.

The OECD QOL metric 'life satisfaction' is derived from the Gallup World Poll, which interviews 1000 residents per country and consists of more than 100 questions on various topics. The metric is a weighted sum of a subset of questions in which the interviewee evaluates

**Table 1. Summary of quality of life metrics across all countries and those featured in the manuscript.**

| Metric | | All Countries | | Featured Countries | |
|---|---|---|---|---|---|
| | Scale | Mean | SD | Mean | SD |
| self-reported health | 0–100 | 67.5 | 13.8 | 73.4 | 9.1 |
| life satisfaction | 0–10 | 6.5 | 0.8 | 6.6 | 0.7 |
| Gini index | 0–1 | 0.32 | 0.05 | 0.31 | 0.04 |
| percentage of users seeking health information | 0–100 | 57.8 | 12.6 | 73.4 | 9.1 |

their current life relative to their best and worst possible lives. Responses are on a scale from 0 to 10, using the Cantril Ladder, with the QOL summary metric also on this scale [19].

We took the Gini index, which represents the income inequality in a country, from the OECD Income Inequality Database [20]. We chose the last year for which a value existed in early 2022 for all featured countries. The year chosen for each country can be found in the OECD data we collate (see Data Availability). The Gini index measures the non-linearity between the proportion of a country's total income cumulatively earned by the bottom $x$% of the population as $x$ increases. If all incomes are equivalent, the relationship is linear, e.g., the "bottom" 20% of the population generates 20% of the total income. The metric ranges from 0–1, 0 meaning high inequality and 1 indicating high equality.

Lastly, we selected the 'percentage of users seeking health information' from the OECD Information and Communications Technology (ICT) dataset on Access and Usage by Households and Individuals [21]. Participants, aged 16 to 74, were asked if they "had used the Internet to seek health information in the last three months". The response was binary, from which a percentage was calculated. In no year were all featured countries measured; as such, we took the most recent value available for each.

We give Table 1, in which we provide descriptive numbers for each metric. For each, we report the scale, the mean, and the standard deviation; the latter two with regard to all countries present in the original dataset and only those featured in our study. For the 'percentage of users seeking health information', the statistics given are based on 2021. Further, we give in S1 Table the values for each QOL metric for each country featured in our study. Lastly, the OECD does not report the number of participants these values are based upon.

## Correlation analysis

The Kolmogorov-Smirnov test [22] (KS-test) was used to determine the normality of our data. Finding it non-normal, we calculated the Spearman's Rank Correlation Coefficients (SRCC) for each country between the COVID-19 cases and the Symptoma risk assessment. The SRCC for two variables, $X$ and $Y$, is defined as

$$SRCC(X, Y) = 1 - \frac{6\Sigma d_i^2}{n(n^2 - 1)},$$

where

$$d_i = R(X_i) - R(Y_i),$$

which is the difference in rank at position $i$, and $n$ is the sample size.

To investigate the direction of effect between COVID-19 cases and Symptoma risk assessment, we performed a lagged-correlation analysis. For two time series $x_t$ and $y_t$, the lagged correlation (LC) was calculated as

$$LC(x_t, y_t, \Delta) = SRCC(x_t, y_{t+\Delta})$$

where Δ represents the lag introduced between the time series. We calculated the LC for Δ ranging from +/-25 days and noted the maximum value of LC obtained in this window and at which value of Δ this occurred. Comparing the maximum LC to the non-lagged SRCC (Δ = 0) indicates the magnitude of the temporal effect, while the value of sΔ indicates direction.

## Results and discussion

**Correlation.**   To quantify the relationship between the COVID-19 cases and the Symptoma risk assessment, we calculated the SRCC between these time series for each country in our dataset. Fig 1A and 1B give examples of the compared time series for Germany and the United States, respectively. The correlation for Germany and United States was 0.733 and -0.187, respectively; the highest and lowest observed correlation seen across all featured countries. The correlations for all 18 countries are given in S1 Table. We find that 15 of the 18 countries (83%) show a positive correlation, while only three show either a negative or no correlation. These three are Sweden, Australia, and the United States. The average correlation across all countries was 0.342 (median = 0.353).

The observed country-specific correlations may reflect different societal cultures. For example, online health resources may be more prevalent in countries with low health care access. In Fig 1C–1F, we show the country-specific correlations against various QOL measures; the GINI-Index (income inequality), self-reported health (self-evaluation of current health), life satisfaction (self-evaluation of current life relative to the best and worst possible lives for them), and the percentage of Internet usage for health-related topics. For each QOL metric, the regression line is given to visually indicate the association's strength with the correlations. The adjusted $R^2$ values were 0.124 (p-value = 0.335), 0.409 (p-value = 0.01), 0.102 (p-value = 0.424) and -0.057 (p-value = 3.117) for Gini-Index, self-reported health, life satisfaction, and the percentage of Internet usage for health-related topics respectively. As such, self-reported health, a measure of how a given population assess their health status, was statistically reflected in the correlations. The relationship was inverse, i.e., a poor estimate of health status indicates a higher correlation between Symptoma risk assessments and COVID-19 cases.

## Correlation shift

In the previous section, we showed that the official COVID-19 case numbers and the Symptoma risk assessment agree well for many countries. Further, this agreement is higher if the self-reported health of the population is low. This yields the question of which variable is time-wise ahead of the other? Do more new COVID-19 cases lead to more persons assessed as high risk by Symptoma, or are already more persons assessed as high risk by Symptoma before the official COVID-19 cases increase? We performed a lagged-correlation analysis whereby the two time series for a given country are shifted relative to each other, and the change in correlation was measured. We note the optimal shift for each country, that being the lag with which the highest correlation is obtained, alongside the change in correlation from the correlation measured with no lag. The change in correlation gives insight into the magnitude of the temporal effect, while the optimal lag indicates direction.

We give all optimal lags, the maximised correlations, and the change with respect to non-lagged correlations in S1 Table. Fig 2 shows a scatter plot highlighting 12 countries whose optimal shift falls in the closed interval [–15, 15]. We chose this constraint to remove six countries whose best shift was directly at the edge of our measured window of [–25, 25] (see S1 Table). When considering all countries in our dataset, the median of the optimal shifts is slightly positive (median: +0.5 days over all 18 countries). When we remove those at the edge of the window, the positivity increases (median: +3 days overall 12 countries). This result suggests that,

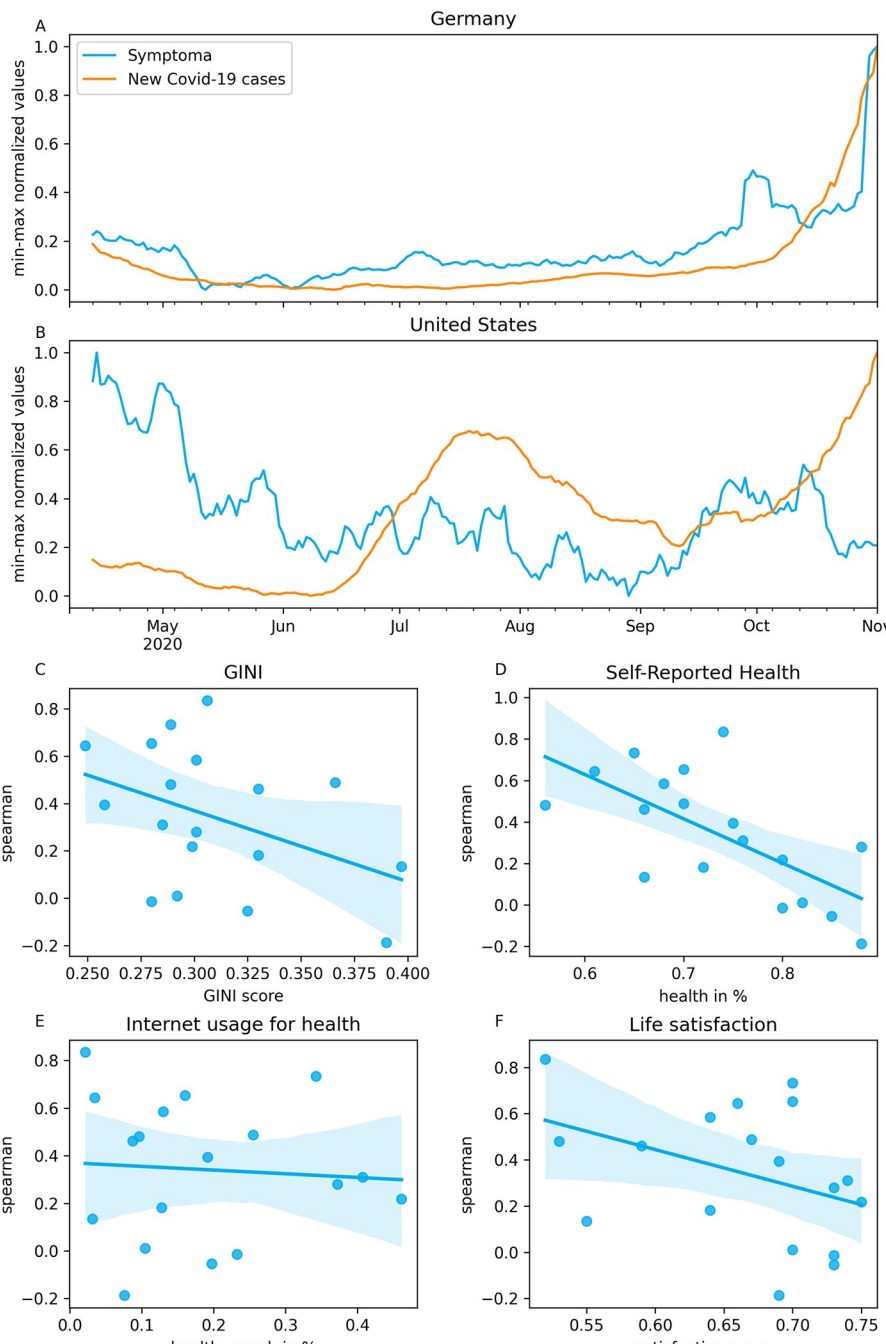

**Fig 1. Correlation analysis between COVID-19 cases and Symptoma risk assessment.** (A), (B) Comparison of COVID-19 cases and Symptoma risk assessment over time for Germany and the United States. (C)—(F) Scatterplot between the spearman correlation coefficients and various QOL measures per country.

on average, the trends observed in the number of users assessed by Symptoma to have a high risk of being infected with COVID-19 occur prior to the same behaviour appearing in the official number of cases. By introducing shifts, the mean SRCC increases from 0.342 (median: 0.353) to 0.411 (median: 0.419).

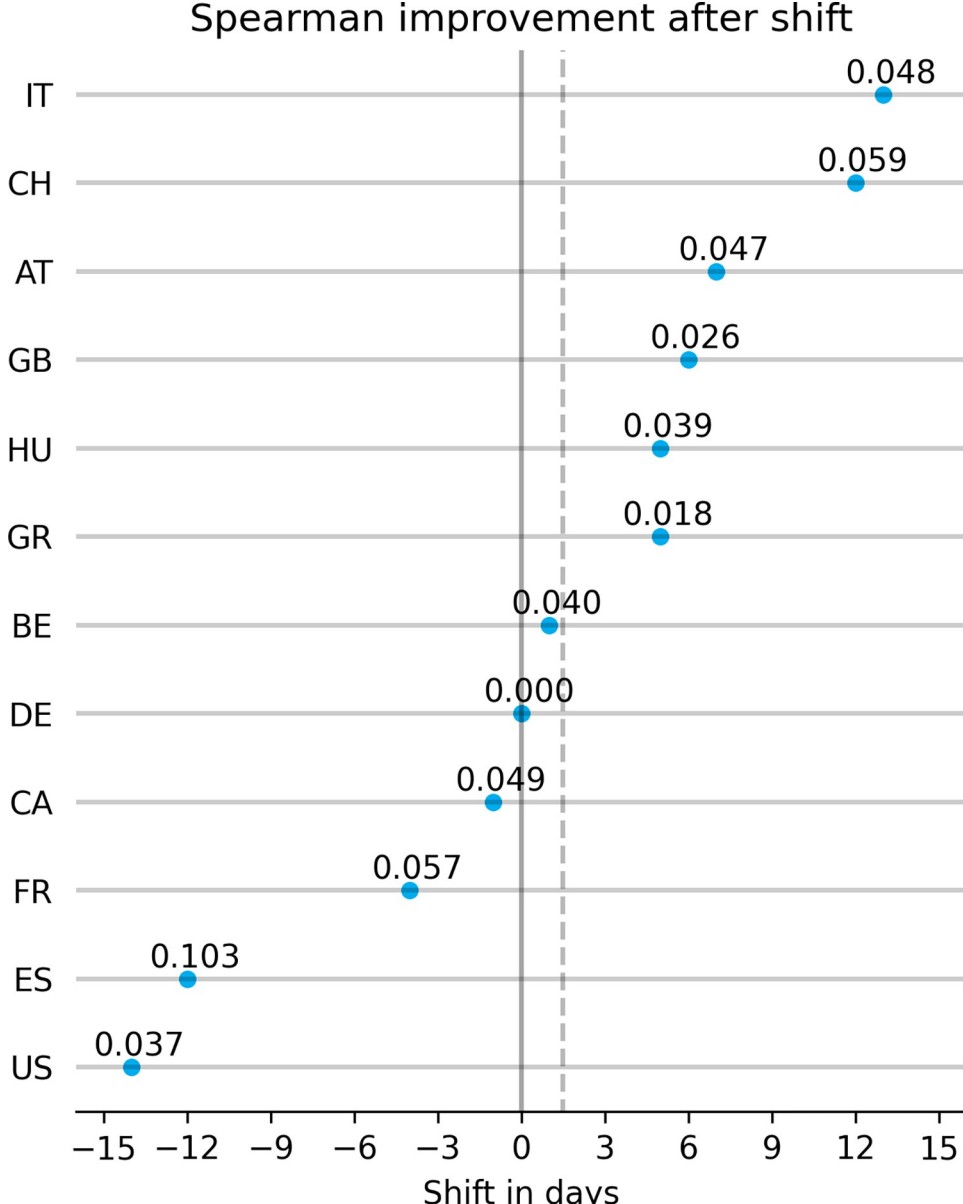

**Fig 2. Shift correlation analysis between COVID-19 cases and Symptoma risk assessment.** The x-axis shows the days the Symptoma time series shifted to reach the maximum correlation. The numbers above the markers show the improvement in correlation with the introduction of a shift. Only countries with a shift within the range [−15, 15] were included in this figure. The dashed grey line represents the mean at 1.5 days.

## Discussion

Above, we have shown that users interacting with an online health resource, a symptom checker, can reflect trends in the COVID-19 progression experienced by a country. The average correlation across the 18 countries featured was 0.342. The remaining variance could be due to the populations behind the time series being inherently different. Namely, the official COVID-19 case numbers only cover the part of the population that got a positive PCR test result. People that were infected but did not get tested are excluded. Symptom checkers provide a risk assessment based on the information provided by the user. A high-risk assessment

does not imply a positive test result. Furthermore, while we found that most countries have a positive correlation between the time series, three countries negatively correlated; Sweden, Australia, and the United States. Of those, Sweden took a different approach to COVID-19 management, preferring minimal lockdowns [23], while the USA reported COVID-19 numbers have been questioned [24, 25].

To understand what may cause the variation in the correlation between the Symptoma risk assessment and the COVID-19 case numbers across countries, we related the country-wise correlations with various QOL metrics. For example, we theorized a relationship with the health-related usage of the internet because people who search for health-related information online are likely to be interested in using an online symptom checker. However, of the four QOL metrics investigated, only self-reported health, a measure of how a given population assess their health status, was significantly negatively associated with the correlations. That means a lower estimate of one's health status across a given population is related to the chatbot better capturing the population-level trends in COVID-19 infections. A lower estimate of one's health could drive one to seek available online resources; however, such an explanation is rejected due to the lack of association with the QOL metric "the percentage of Internet usage for health-related topics". Similarly, we saw no relationship between the correlations and a country's income equality or life satisfaction. From this, it is still unclear what drives the difference: some countries exhibit a high correlation between the measured time series while others do not.

Lastly, we found that, on average, the trends observed in the number of users assessed by Symptoma to have a high risk of being infected with COVID-19 occur prior to the same behaviour appearing in the official number of cases. Specifically, we found the best overlap between the time series when introducing an average delay of three days, this change increasing the mean SRCC from 0.342 to 0.411. This relationship could be due to the delay between the initial onset of symptoms, the most likely time to use an online symptom checker, and getting an official positive test result. For most countries, for the average positive case, there is a delay between the following events: contact with an infected individual, the onset of symptoms, administration of a PCR test, and official record of the PCR test result. In an extreme example of such a scenario, a positive assessment from the Symptoma symptom checker, the chatbot featured in this study, was even required to obtain a PCR test from the Viennese government [26]. Given this, a delay was inevitable between the positive assessment and a positive PCR test in Vienna.

While the correlations increased universally with the introduction of a shift, for some countries, the optimal shift was negative, i.e., the COVID-19 pandemic progression precedes the trends observed in the chatbot. For example, for Spain, which exhibited the largest increase in correlation (+0.103), the optimal shift was -12 days. We believe this is due to the number of recorded cases in Spain being almost monotonic over the period analysed. Other countries experience an initial peak followed by a Summer season with fewer infections, a time-series feature with which the lagged correlation can attempt to align. For Spain, infection numbers were instead stable after the initial onset. The absence of an initial peak was likely due to limited testing capacity when Spain was one of the epicentres of the pandemic's first wave in Europe [27].

## Conclusion

In this paper, we investigated how collated user interactions with the online symptom checker Symptoma can be related to the progression of the worldwide COVID-19 pandemic. We found that, on average, there is a positive correlation between global COVID-19 cases and

Symptoma users assessed with a high risk of being infected with COVID-19. Further, we found a relationship between said correlation and the self-reported health of a country's population. Lastly, we found that, on average, the number of Symptoma users assessed as having a high risk of being infected with COVID-19 increased/decreased before the global COVID-19 cases increased/decreased. While further work is required, given these results, it is clear that symptom checkers could help detect coronavirus hot spots early on and be a crucial resource in future pandemic responses.

## Supporting information

**S1 Table. QOL metrics, correlations, and shifted correlations.**
(DOCX)

## Author Contributions

**Conceptualization:** Bernhard Knapp, Jama Nateqi.

**Data curation:** Marc Zobel.

**Formal analysis:** Marc Zobel, Alistair Martin.

**Funding acquisition:** Jama Nateqi.

**Investigation:** Marc Zobel, Alistair Martin.

**Methodology:** Marc Zobel, Alistair Martin.

**Project administration:** Bernhard Knapp, Jama Nateqi.

**Resources:** Jama Nateqi.

**Software:** Marc Zobel.

**Supervision:** Bernhard Knapp.

**Validation:** Marc Zobel, Alistair Martin.

**Visualization:** Marc Zobel, Bernhard Knapp, Alistair Martin.

**Writing – original draft:** Marc Zobel.

**Writing – review & editing:** Marc Zobel, Bernhard Knapp, Jama Nateqi, Alistair Martin.

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
