## [Decision Letter · Decision Letter 0]

26 Oct 2021

PONE-D-21-08501Predicting global trends in COVID-19 cases via online symptom checker self-assessmentsPLOS ONE

Dear Dr. Zobel,

Thank you for submitting your manuscript to PLOS ONE. After careful consideration, we feel that it has merit but does not fully meet PLOS ONE’s publication criteria as it currently stands. Therefore, we invite you to submit a revised version of the manuscript that addresses the points raised during the review process.

Please note that the Academic Editor has requested the figures in a higher resolution. However, PLOS staff has checked the resolution and there is no need to upload higher resolution images in your revision, as the original resolution, which is sufficient, will be retained if the manuscript is published.

We look forward to receiving your revised manuscript.

Kind regards,

Hanna Landenmark

Senior Editor

PLOS ONE

on behalf of 

Thomas Martin Deserno,

Journal Requirements:

https://journals.plos.org/plosone/s/file?id=wjVg/PLOSOne_formatting_sample_main_body.pdf andhttps://journals.plos.org/plosone/s/file?id=ba62/PLOSOne_formatting_sample_title_authors_affiliations.pdf2. Thank you for stating the following in the Financial Disclosure section:

[This study has received funding from the European Union’s Horizon 2020 research and innovation programme under grant agreement No 830017 and by the Austrian Research Promotion Agency under grant agreement No 880939 (supported by the Federal Ministries Republic of Austria for Digital and Economic Affairs and Climate Action, Environment, Energy, Mobility, Innovation and Technology).].   

We note that one or more of the authors are employed by a commercial company: Symptoma GmbH  1.
Please provide an amended Funding Statement declaring this commercial affiliation, as well as a statement regarding the Role of Funders in your study. If the funding organization did not play a role in the study design, data collection and analysis, decision to publish, or preparation of the manuscript and only provided financial support in the form of authors' salaries and/or research materials, please review your statements relating to the author contributions, and ensure you have specifically and accurately indicated the role(s) that these authors had in your study. You can update author roles in the Author Contributions section of the online submission form. Please also include the following statement within your amended Funding Statement.“The funder provided support in the form of salaries for authors [insert relevant initials], but did not have any additional role in the study design, data collection and analysis, decision to publish, or preparation of the manuscript. The specific roles of these authors are articulated in the ‘author contributions’ section.” If your commercial affiliation did play a role in your study, please state and explain this role within your updated Funding Statement.     2. Please also provide an updated Competing Interests Statement declaring this commercial affiliation along with any other relevant declarations relating to employment, consultancy, patents, products in development, or marketed products, etc.   Within your Competing Interests Statement, please confirm that this commercial affiliation does not alter your adherence to all PLOS ONE policies on sharing data and materials by including the following statement: "This does not alter our adherence to  PLOS ONE policies on sharing data and materials.” (as detailed online in our guide for authors http://journals.plos.org/plosone/s/competing-interests) . If this adherence statement is not accurate and  there are restrictions on sharing of data and/or materials, please state these. Please note that we cannot proceed with consideration of your article until this information has been declared. Please include both an updated Funding Statement and Competing Interests Statement in your cover letter. We will change the online submission form on your behalf. Please know it is PLOS ONE policy for corresponding authors to declare, on behalf of all authors, all potential competing interests for the purposes of transparency. PLOS defines a competing interest as anything that interferes with, or could reasonably be perceived as interfering with, the full and objective presentation, peer review, editorial decision-making, or publication of research or non-research articles submitted to one of the journals. Competing interests can be financial or non-financial, professional, or personal. Competing interests can arise in relationship to an organization or another person. Please follow this link to our website for more details on competing interests: http://journals.plos.org/plosone/s/competing-interests 3. We note that you have indicated that data from this study are available upon request. PLOS only allows data to be available upon request if there are legal or ethical restrictions on sharing data publicly. For information on unacceptable data access restrictions, please see http://journals.plos.org/plosone/s/data-availability#loc-unacceptable-data-access-restrictions.In your revised cover letter, please address the following prompts:a) If there are ethical or legal restrictions on sharing a de-identified data set, please explain them in detail (e.g., data contain potentially identifying or sensitive patient information) and who has imposed them (e.g., an ethics committee). Please also provide contact information for a data access committee, ethics committee, or other institutional body to which data requests may be sent.b) If there are no restrictions, please upload the minimal anonymized data set necessary to replicate your study findings as either Supporting Information files or to a stable, public repository and provide us with the relevant URLs, DOIs, or accession numbers. Please see http://www.bmj.com/content/340/bmj.c181.long for guidelines on how to de-identify and prepare clinical data for publication. For a list of acceptable repositories, please see http://journals.plos.org/plosone/s/data-availability#loc-recommended-repositories.We will update your Data Availability statement on your behalf to reflect the information you provide.

Additional Editor Comments :

Please provide the figures in a better quality before I can send the yrticle out for review. They are unreadable.

Reviewers' comments:

Reviewer's Responses to Questions

**Comments to the Author**

1. Is the manuscript technically sound, and do the data support the conclusions?

Reviewer #1: No

Reviewer #2: Partly

2. Has the statistical analysis been performed appropriately and rigorously? 

Reviewer #1: No

Reviewer #2: No

3. Have the authors made all data underlying the findings in their manuscript fully available?

Reviewer #1: No

Reviewer #2: No

4. Is the manuscript presented in an intelligible fashion and written in standard English?

Reviewer #1: Yes

Reviewer #2: No

5. Review Comments to the Author

Reviewer #1: This an interesting study to investigate the correlation between the data of symptom checkers by those who are either affected by Covid 19 or are suspected to this disease, and the data of cases according to the region and time. However, this analysis need more improvement and the results have not been presented properly. Here there are some comments:

1- in the method, the normal or not normal distribution of the variables are required to be checked by test such as Kolmogorov–Smirnov (k-s) test and then the PCC to be used, Spearman coloration coefficient might be the suitable option if pvalue of K-S test >0.05. We need to check the data distribution and then the test can be chosen.

2- There is the lack of section to introduce the available data before analysis. Both variables according to the time and geography need to be summarized using descriptive indexes and the after having insight regarding the data , we need to progress for data analysis.

3- How was the proportion of users of symptom checkers based on time and geography ? in case of the balanced data availability, the result can be promising and we can compare the results for users with the same country. We need to make sure that the sampling has been done correctly with enough cases for each country as we use the data of positive cases according to the country (John's Hopkin's Uni data ). More clarification in methodology is really required.

We need to make sure that we have had population in both variables with enough numbers in each cluster in which each cluster is for a country.

4- in discussion, we need to justify this controversy that the positive cases in each country are those whose PCR test is positive, However, the symptom checkers might be those who are positive of COVID 19, but they did not do the PCR test. Thus, the study population of the two variables might be different.

5- More information regarding the topic is required for discussion to compare the result with the other works and interpret the findings. The discussion and conclusion need update and improvement.

Reviewer #2: Thank you for sending this novel piece of work.

I like the idea and effort behind the manuscript.

Few are my recommendations and suggestions:

1.The methodology of the paper should be succinctly stated. Like study design, study setting. I do not see any details for so.

2.I see that authors emphasized in the the abstract and write up on digital solutions, but talk briefly on the digital health approaches .

3.The objectives stated do not deem aligned with the title, methods and overall manuscript , would suggest revisiting the objectives.

4.Very little is explained in the section of analysis, I miss a logical sequence. As mentioned, the concepts do not seem to be aligned. This can be made more smooth and reader understandable.

5.The authors should give a sound rationale to explain the link between the type of analyses and a justification for doing such analysis. In my opinion there could have been more robust analysis(regression)to do so.

6.I would suggest that the authors also categorize the countries region/ income status wise and then explain the results, figures are very hap hazard and confusing, have a lot more room for improvement

6. PLOS authors have the option to publish the peer review history of their article (what does this mean?). If published, this will include your full peer review and any attached files.

Reviewer #1: No

Reviewer #2: **Yes: **Dr.Maleeha Naseem

---

## [Author Response · Author response to Decision Letter 0]

20 Jan 2022

Dear Reviewers,

On behalf of all the authors of the paper, we would like to thank you for the in-depth review. The suggested changes have been mostly implemented after much careful consideration. In the following sections, we describe in detail how each comment was addressed:

Reviewer 1:

1- in the method, the normal or not normal distribution of the variables are required to be checked by test such as Kolmogorov–Smirnov (k-s) test and then the PCC to be used, Spearman correlation coefficient might be the suitable option if pvalue of K-S test >0.05. We need to check the data distribution and then the test can be chosen.

As suggested, we tested the normality of each time series via the Kolmogorov-Smirnov test. We failed to reject the null hypothesis for most time series, the KS statistic indicating that they are non-normal (p-value > 0.05). Given this, we replaced the analyses based on the Pearson correlation coefficients throughout with the Spearman rank correlation coefficients. Previous observed trends and conclusions drawn from them were, in general, unaffected by this change.

Reviewer 1:

2- There is the lack of section to introduce the available data before analysis. Both variables according to the time and geography need to be summarized using descriptive indexes and the after having insight regarding the data , we need to progress for data analysis.

We improved our Data section by expanding the description of the time frame, while also providing the count of the countries and the number of days in total. The list of countries included in this study is also now given in full. 

Only countries with consecutive data, that being no days without a single high-risk assessment given, between 7th of April and 1st of November 2020 were included in the further analysis. This results in a dataset of 18 countries with 208 days in each. The respective countries are: Australia, Austria, Belgium, Canada, Czech Republic, France, Germany, Greece, Hungary, Ireland, Italy, Netherlands, Spain, Sweden, Switzerland, Turkey, United Kingdom, United States. Within the text, we refer to this data as ‘Symptoma risk assessments’.

Alongside the above text, we also inserted additional instances of reference to our previous work which give an overview of how our risk assessments are performed.

Reviewer 1:

3- How was the proportion of users of symptom checkers based on time and geography? in case of the balanced data availability, the result can be promising and we can compare the results for users with the same country. We need to make sure that the sampling has been done correctly with enough cases for each country as we use the data of positive cases according to the country (John's Hopkin's Uni data ). More clarification in methodology is really required. We need to make sure that we have had population in both variables with enough numbers in each cluster in which each cluster is for a country.

We thank the reviewer for their concerns. Since Symptoma and to our knowledge any other symptom checker still do not have the same reach as a country-wide testing strategy, unbalanced data was always expected. We filtered out countries, which do not show consecutive data in our time frame:

Only countries with consecutive data, that being no days without a single high-risk assessment given, between 7th of April and 1st of November 2020 were included in the further analysis.

Additionally, we scaled the data to adjust for different magnitudes:

Additionally, we scaled the data so that zero equals the minimum and one equals the maximum. This was done to make the time-series comparable visually and to protect the sensitive user numbers.

Reviewer 1:

4- in discussion, we need to justify this controversy that the positive cases in each country are those whose PCR test is positive, However, the symptom checkers might be those who are positive of COVID 19, but they did not do the PCR test. Thus, the study population of the two variables might be different. 

We strongly agree that there are underlying key differences in the populations reflected by the two data sources used within our manuscript. For example, there are clear weekly seasonality patterns in the data provided by Johns Hopkins. Our preprocessing steps, namely, applying a smoothing kernel and min-max scaling addresses this somewhat. However, we do note that this does not solve potential differences in the sampling population. We have added the following to our discussion to reflect the reviewer’s valid concern: 

Note that the official COVID-19 case numbers only cover the part of the population that got a positive PCR test result. This excludes people that are infected but did not get tested. Symptom checkers provide a risk assessment based on the information provided by the user. A high risk assessment does not imply a positive test result. This leads to the two populations of our comparison being different.

Reviewer 1:

5- More information regarding the topic is required for discussion to compare the result with the other works and interpret the findings. The discussion and conclusion need update and improvement.

Reviewer 2:

1.The methodology of the paper should be succinctly stated. Like study design, study setting. I do not see any details for so.

We adjusted the abstract, introduction and objectives to reflect our methodology more succinctly. Our objectives now state:

Objectives

Assess the correlation between COVID-19 risk assessments from an online symptom checker and current trends in COVID-19 infections and investigate the relationship between those correlations and four quality of life measures.

Further, we added the following section to the end of the introduction to provide an additional overview.

With regards to the COVID-19 pandemic, we investigate whether the lab-confirmed numbers of new COVID-19 cases correlate with the number of high COVID-19 risk assessments given by the ‘Symptoma’ symptom checker. Similar correlation analysis were performed by others in the context of temperature [7,8], air pollution [9], and income inequality [10]. We show that our hypothesis holds for a majority of countries, the count of high-risk assessments reflecting the trends in COVID-19 cases nationally. Further, we relate these country-wise correlations with 4 different socio-economic factors and find a significant connection to self-reported health from the OECD Better Life Index 2017. Lastly, we investigate the optimal lag between the official case counts and the symptom checker assessments to maximise correlations to indicate whether the assessments could be predictive of the pandemic progression.

Reviewer 2:

2.I see that authors emphasized in the the abstract and write up on digital solutions, but talk briefly on the digital health approaches . 

We thank the reviewer for the input. We expanded upon our previous introduction to give a broader range of application context and examples of digital health solutions: 

Other surveys list the applications as screening, triage, diagnosis, monitoring and contact tracing [4]. A prominent example of contact tracing software is the mobile app TraceTogether used in Singapore [5] to track past contacts of infected individuals. Surveillance as the continuous, systematic collection, analysis and interpretation of health-related data needed for the planning, implementation, and evaluation of public health practice [4] is the goal of Online surveillance-mapping tools like SORMAS[6] and HealthMap[7]. Monitoring systems are usually deployed in a more private context, to record the condition of already infected individuals. The tool Sense FollowUp app[8] has been used to monitor Ebola patients remotely and raise an alarm in case of increased temperature readings. One specific type of application in the context of triage is online symptom checkers which are experiencing an increase in popularity [9].

Reviewer 2:

3.The objectives stated do not deem aligned with the title, methods and overall manuscript , would suggest revisiting the objectives. 

We agree that the focus of the paper was not reflected accurately by the objectives. To address this, the title “Predicting global trends in COVID-19 cases with online symptom checker self-assessments” has been revised to “Correlating global trends in COVID-19 cases with online symptom checker self-assessments” to better reflect our methodology. Likewise, the objectives were changed to:

Assess the correlation between COVID-19 risk assessments from an online symptom checker and current trends in COVID-19 infections and investigate the relationship between those correlations and four quality of life measures.

which reflects better the investigation of the correlation of symptom checker user assessments and global COVID-19 cases. 

Reviewer 2:

4.Very little is explained in the section of analysis, I miss a logical sequence. As mentioned, the concepts do not seem to be aligned. This can be made more smooth and reader understandable.

We improved our methods and discussion section to improve the logical sequence. More insight was given to show how the data was generated and underlying goals. The list of countries included in our data set got added to the data section:

This results in a dataset of 18 countries with 208 days in each. The respective countries are: Australia, Austria, Belgium, Canada, Czech Republic, France, Germany, Greece, Hungary, Ireland, Italy, Netherlands, Spain, Sweden, Switzerland, Turkey, United Kingdom, United States. Within the text, we refer to this data as ‘Symptoma risk assessments’.

Pre-processing

Delayed reporting of new COVID-19 cases is a well-known problem [12]. To account for this problem and remove unnatural spikes (e.g., weekends), we applied a seven-day mean filter to the COVID-19 cases as well as the Symptoma risk assessment. Additionally, we scaled the data so that zero equals the minimum and one equals the maximum. This was done to make the time-series comparable visually and to protect the sensitive user numbers.

Additionally, our discussion has been expanded to include the quality of life measures suggested in another comment below.

On the y-axis of Figure 1C-F, we show in addition the QOL measures GINI-Index, self-reported health, life satisfaction, and the percentage of Internet usage for health-related topics. Hence, Figure 1C-F is a scatter plot between those four QOL measures and ‘SRCCs between new COVID-19 positive cases and Symptoma high-risk cases’. A linear model was fit to the data of each Subfigure 1C-F and a regression line was drawn to illustrate the result.

We theorized a relationship between health-related usage of the internet SRCC, because people who search for health-related information on the internet are likely to interested in symptom checker results as well. Figure 1E shows no correlation between health-related usage and SRCC. The regression line reflects that with a slope close to 0 (-0.159) and an R² of 0.155.

Our reason for choosing the GINI-Index was that the low-income population should be more inclined to use free online services. Thus we theorize that countries with higher inequality would have a larger portion of the low-income population, which would be more inclined to use Symptoma. Figures 1C and 1F show a weak inverse correlation between SRCC and the QOL measures GINI-Index (SRCC: -0.331) and life satisfaction (SRCC: -0.378). The respective p-values (0.084 and 0.779) provided by the linear model show no statistical significance for this relationship. Figure 1D shows a strong negative correlation between SRCC and self-reported health (SRCC: -0.665). The result is supported by the linear model (R²: 0.443, p-value: 0.003) and is statistically significant.

Lastly, we streamlined our discussion of the correlation shift analysis to better explain the setting and goals of this analysis:

Correlation shift

In the previous section, we showed that there is good agreement between new COVID-19 cases and Symptoma risk assessment for most countries. This yields the question of which variable is time-wise ahead of the other. Do more new COVID-19 cases lead to more persons assessed as high risk by Symptoma or are already more persons assessed as high risk by Symptoma before the official COVID-19 cases increase? To shed light on this question we performed a lagged-correlation analysis. In Figure 2, we show a scatter plot highlighting 12 countries whose optimal shift falls in the closed interval [ -15, 15 ]. We chose this constraint to remove 6 countries whose best shift was directly at the edge of our window of [ -25, 25 ]. For the majority of countries (n: 18) the optimal shift is positive (median: +0.5 days overall 18 countries). The same holds for the filtered set of countries (median: +3 days overall 12 countries). This means that on average the number of users assessed by Symptoma to have a high risk of being infected with COVID-19 increases/decreases before the official number of cases increases/decreases. By introducing shifts, the mean SRCC increases from 0.342 (median: 0.353) to 0.411 (median: 0.419).

Reviewer 2:

5.The authors should give a sound rationale to explain the link between the type of analyses and a justification for doing such analysis. In my opinion there could have been more robust analysis(regression)to do so.

We thank the reviewer for their comment. The key analysis of the manuscript is the correlation shift analysis. We hypothesised that this analysis would give insight as to whether user-interaction with a symptom checker can reflect a pandemic’s progression. Should this be the case, and as commented on within the manuscript, then further work invested into predictive modelling and forecasting would be planned. As mentioned above, we have added additional text to both the Abstract, Introduction and Methods to help clarify our objectives.

Reviewer 2:

6. I would suggest that the authors also categorize the countries region/ income status wise and then explain the results

We followed the reviewer’s suggestion for a great addition to our manuscript. We selected four relevant Quality of Life (QoL) metrics and tested them for association with our previous results. Specifically, from the OECD Better Life Index 2017 (BLI 2017) we selected “self-reported health” and “life satisfaction”. the GINI Index, which represents income inequality, and, lastly, the “percentage of internet users seeking health information”. We assessed the relationship of those measures with the correlation between Symptoma users assessed with a high risk for COVID-19 and the global COVID-19 cases provided by Johns Hopkins University. We found a significant negative relationship between self-reported health and the correlation between Symptoma and global COVID-19 cases (SRCC: -0.665, R²: 0.443, adj. p-value: 0.003).

Reviewer 2:

figures are very hap hazard and confusing, have a lot more room for improvement

We are sorry to hear that the presentation of data was confusing. In light of this comment, we have made the following figure alterations. Figure 1 was changed to show the normalized time series of the Symptoma assessment and the COVID-19 cases for a few example country in subfigures A & B. In subfigures C-F, the relationship between the above QoL measures and the calculated SRCC between Symptoma and new COVID-19 cases is given. Figure 2 was reworked entirely to remove unnecessary information; now only the core information of the shifted correlation analysis is shown. Namely, only the optimal lag per country is presented. Further, the countries at the edge of our shift window of [ -25, 25 ] were removed. We believe that with all of the above alterations, figure clarity is greatly enhanced.

Lastly, we would again like to thank the reviewers for their comments. As outlined above, we have greatly improved the manuscript based on their feedback.

Yours Sincerely,

Marc Zobel

Alistair Martin

Jama Nateqi

Bernhard Knapp

---

## [Decision Letter · Decision Letter 1]

2 Aug 2022

PONE-D-21-08501R1Correlating global trends in COVID-19 cases with online symptom checker self-assessmentsPLOS ONE

Dear Dr. Zobel,

Thank you for submitting your manuscript to PLOS ONE. After careful consideration, we feel that it has merit but does not fully meet PLOS ONE’s publication criteria as it currently stands. Therefore, we invite you to submit a revised version of the manuscript that addresses the points raised during the review process.

We look forward to receiving your revised manuscript.

Kind regards,

Thomas Martin Deserno, Ph.D.

Academic Editor

PLOS ONE

Reviewers' comments:

Reviewer's Responses to Questions

**Comments to the Author**

1. If the authors have adequately addressed your comments raised in a previous round of review and you feel that this manuscript is now acceptable for publication, you may indicate that here to bypass the “Comments to the Author” section, enter your conflict of interest statement in the “Confidential to Editor” section, and submit your "Accept" recommendation.

Reviewer #1: All comments have been addressed

Reviewer #3: (No Response)

2. Is the manuscript technically sound, and do the data support the conclusions?

Reviewer #1: Yes

Reviewer #3: Partly

3. Has the statistical analysis been performed appropriately and rigorously? 

Reviewer #1: Yes

Reviewer #3: No

4. Have the authors made all data underlying the findings in their manuscript fully available?

Reviewer #1: Yes

Reviewer #3: No

5. Is the manuscript presented in an intelligible fashion and written in standard English?

Reviewer #1: Yes

Reviewer #3: Yes

6. Review Comments to the Author

Reviewer #1: As all the comments are considered and responded, the study improved and well enough for publication in the PLOS One Journal.

Reviewer #3: proportion of users of symptom checkers based on time/geography: Still no details on the total numbers to be able to compare between countries, since this is relevant information - though you scaled the data this would be needed, at least as appendix.

Methods:

lack of the section to introduce the data: Thanks for enhancing this section. It would be beneficial to provide the full data set as supplement file or refer to an externally available repository that contains the data. Still missing is the absolute numbers of each data set. Especially for the Quality of life measures from the OECD a detailed description is missing. Also descriptive numbers for all data sets used for the paper are required to fully understand it. It is important to see the 4 QoL metrics per country in detail.

the correlation analysis description needs further rework, since it is still not comprehensible how you exactly did that "lagged-correlation" analysis. It remains unclear why the authors team has chosen Germany and US to include in Figure 1. Also it would be good to not only introduce the mean/median correlation of all included countries but instead add it for each country (e.g. in a table)

Results + Discussion: This section mixes up results and discussion. You need to strictly divide between the results based on your methods and then discuss them. Currently you mix results and discussion. Also the sequence is still confusing here. Still needs rework here. Confidence interval is missing for the regression line. Correlation shift results section needs further improvement. The correlation improvement by country is very small and without the absolute value of the correlation for each country without the shift not useful. Conspicuous is the largest increase of 0.103 for Spain, that you unfortunately did not take into account for any discussion.

Figures: still need rework, Figure 1 (A) + (B): x-axis labels missing, Figure 1 (C) - (F): x-axis labels + scale is not explained before in detail, you also decided to cut x-axis e.g. GINI score was introduced from 0 - 1 but not reflected in the Figure. You mention the 4 correlations 0 or smaller in the text for GINI but not this other plots (D), (E), (F)

7. PLOS authors have the option to publish the peer review history of their article (what does this mean?). If published, this will include your full peer review and any attached files.

Reviewer #1: No

Reviewer #3: No

---

## [Author Response · Author response to Decision Letter 1]

25 Nov 2022

Dear Reviewers,

On behalf of all the authors of the paper, we would like to thank you for the in -depth review.

We have implemented most of the suggested changes after much careful consideration. In

the following sections, we describe in detail how we addressed each comment:

Methods:

lack of the section to introduce the data: Thanks for enhancing this section. It would be

beneficial to provide the full data set as supplement file or refer to an externally available

repository that contains the data. Still missing is the absolute numbers of each data set.

We have expanded the Data Availability statement to include a repository link in which the

scaled user count per country per day over the period analysed is stored. Further, we provide

all other data (e.g., QoL metrics) used within our manuscript for complete reproducibility. We

feel strongly that this meets the Minimal Data Set Definition outlined by PLOS; namely, i t

allows the reproduction of all results and figures. In addition to the above, the Data

Availability statement also outlines the process by which a reader can obtain the underlying

untransformed data, i.e., the number of users assessed as “high risk” per day for each country

included in this study. Specifically, the reader must request this data via email. For the review

process, we are happy to supply the data via the editor to the reviewer, should they so wish.

Especially for the Quality of life measures from the OECD a detailed description is missing. Also

descriptive numbers for all data sets used for the paper are required to fully understand it. It is

important to see the 4 QoL metrics per country in detail.

We have expanded the relevant section within Materials & Methods to detail how each

Quality of Life (QoL) metric is derived. Furthermore, the descriptive summary statistics on

these QoL metrics globally, e.g., the mean and standard deviation, are now provided to allow

for context outside the subset of countries featured in our study. Lastly, in the supplementary

information, we now give the exact numbers for the four QoL metrics for the countries

featured in our study.

the correlation analysis description needs further rework, since it is still not comprehensible

how you exactly did that “lagged-correlation” analysis.

We thank the reviewer for their comment. We have expanded the correlation analysis section

to clarify the lagged correlation analysis. Specifically, we now provide a formula such that it is

defined explicitly which should prevent future uncertainty.

It remains unclear why the authors team has chosen Germany and US to include in Figure 1.

Within the text, we stated that these countries are merely examples, one for which there is a

high correlation (Germany) and one for which there is a low correlation (USA). We felt that an

example of the two data sources overlapped provided understanding of the work. However,

we wished not to be accused of cherry-picking by featuring only a good example; thus, we

displayed the two countries mentioned above.

However, and likely the source of the reviewer’s comment, the explanation regarding this

selection, and the associated discussion, was separate from the first introduction of Figure 1.

We suggest that this separation led to the reviewer’s confusion. The relevant text has been

shifted such that we give the context for the country selection alongside the Figure’s first

introduction.

Also it would be good to not only introduce the mean/median correlation of all included

countries but instead add it for each country (e.g. in a table)

We have added the requested table to the supplementary information and included

reference to it within the manuscript.

Results + Discussion:

This section mixes up results and discussion. You need to strictly divide between the results

based on your methods and then discuss them. Currently, you mix results and discussion. Also,

the sequence is still confusing here. Still needs rework here.

We apologise for the lack of clarity here in our previous manuscript. In light of this comment,

we implemented a significant restructuring of the manuscript to make a clear distinction

between our results and their discussion. This restructuring includes introducing a dedicated

discussion section whereby we comment upon the results and what may be driving said

observations.

Confidence interval is missing for the regression line.

We have now added confidence intervals for all figures containing regressions (Figures 1C-

1F). Please note that we had already reported significance values (p-values) for these models

within the text to indicate the goodness of fit.

Correlation shift results section needs further improvement. The correlation improvement by

country is very small and without the absolute value of the correlation for each country without

the shift not useful.

As requested above, we have added a table containing the correlation for each country as a

table within the supplementary information. Further, this table also includes the maximum

correlation obtained when we apply a shift and the shift with which this occurs. We reference

this table within the text, and the information is used throughout the discussion.

Conspicuous is the largest increase of 0.103 for Spain, that you unfortunately did not take into

account for any discussion.

We thank the reviewer for this observation. After investigation, we added the following

discussion to the manuscript:

“While the correlations increased universally with the introduction of a shift, for some

countries, the optimal shift was negative. For example, for Spain, which exhibited the largest

increase in correlation (+0.103), the optimal shift was -12 days, i.e., the COVID-19 pandemic

progression precedes the trends observed in the chatbot. We believe this is due to the

number of recorded cases in Spain being almost monotonic over the period analyzed. Other

countries experience an initial peak followed by a Summer season with fewer infections. For

Spain, infection numbers were instead stable after the initial onset. The absence of an initial

peak was likely due to limited testing capacity when Spain was one of the epicentres of the

pandemic’s first wave in Europe [27].”

Figures: still need rework, Figure 1 (A) + (B): x-axis labels missing,

We have added labels to Figure 1B. As Figure 1A shares an x-axis with Figure 1B, we have

chosen to drop the axis labels as the elements do not add new information to the graphic.

Figure 1 (C) - (F): x-axis labels + scale is not explained before in detail,

As suggested by the reviewer above, we added further explanation of these metrics to our

Materials & Methods section, including a complete description of the respective scales. To

aid the reader, we have also briefly reiterated what each metric represents, when the name is

unclear, within the Results & Discussion.

you also decided to cut x-axis e.g. GINI score was introduced from 0 - 1 but not reflected in the

Figure.

Plotting the complete range of the GINI score and other QoL metrics is unreasonable ,

considering that the subset of countries featured in our study appears only in a small region.

Including everything would lead to an uninformative visualisation; the data is forced

together, even overlapping. However, we understand that limiting the axis to that of the

subset could be used to present a false narrative. We prevent this situation by supplying the

regression for each metric against the correlation, including the significance, which

objectively indicates the strength of relationships to the reader.

You mention the 4 correlations 0 or smaller in the text for GINI but not this other plots (D), (E),

(F)

We are unsure about the reviewer’s comment as the previous iteration of our manuscript did

not contain a discussion about the subset of correlations that were negative and how they

related to any of the QoL metrics, including the Gini index. The negative correlations and

potential reasons why the respective countries diverged from the chatbot trends were

discussed in detail but independently from the QoL metric analysis. Regardless, we hope that

the extensive revision of the results and discussion section, as mentioned above, has

provided additional clarity to this section and resolved the review’s concern.

In addition to the changes based on the reviewer’s comments, we want to draw attention to

a slight correction we made to the manuscript regarding our data pipeline. We stated

previously that we filtered countries which lacked data for even a si ngle day. This was

incorrect. Instead, we filtered countries for which less than 200 of the 208 days had data, i.e.,

more than eight days missing. For reference, eight days is less than 4% of the analysed time

period. Zeros were inserted for days without values. Please note that this had no downstream

effect, as the previous manuscript versions were already based on this filter. Further, during

this review, the manuscript underwent another editing process which resulted in a couple of

small changes. The last notable change is that the author list has been reordered to reflect

changes within our company over the revision period. We hope this change can be

propagated in your system and is not a point of concern.

Lastly, we would again like to thank the reviewers for their comments. As outlined above, we

have greatly improved the manuscript based on their feedback.

Yours Sincerely,

Marc Zobel

Bernhard Knapp

Jama Nateqi

Alistair Martin

---

## [Decision Letter · Decision Letter 2]

31 Jan 2023

Correlating global trends in COVID-19 cases with online symptom checker self-assessments

PONE-D-21-08501R2

Dear Dr. Zobel,

We’re pleased to inform you that your manuscript has been judged scientifically suitable for publication and will be formally accepted for publication once it meets all outstanding technical requirements.

Kind regards,

Sathishkumar V E

Academic Editor

PLOS ONE

Additional Editor Comments (optional):

Reviewers' comments:

Reviewer's Responses to Questions

**Comments to the Author**

1. If the authors have adequately addressed your comments raised in a previous round of review and you feel that this manuscript is now acceptable for publication, you may indicate that here to bypass the “Comments to the Author” section, enter your conflict of interest statement in the “Confidential to Editor” section, and submit your "Accept" recommendation.

Reviewer #1: All comments have been addressed

2. Is the manuscript technically sound, and do the data support the conclusions?

Reviewer #1: Yes

3. Has the statistical analysis been performed appropriately and rigorously? 

Reviewer #1: Yes

4. Have the authors made all data underlying the findings in their manuscript fully available?

Reviewer #1: Yes

5. Is the manuscript presented in an intelligible fashion and written in standard English?

Reviewer #1: Yes

6. Review Comments to the Author

Reviewer #1: As all the comments are considered and responded, the study improved and well enough for publication in the PLOS One Journal.

7. PLOS authors have the option to publish the peer review history of their article (what does this mean?). If published, this will include your full peer review and any attached files.

Reviewer #1: No

---

## [Editor Report · Acceptance letter]

2 Feb 2023

PONE-D-21-08501R2 

Correlating global trends in COVID-19 cases with online symptom checker self-assessments 

Dear Dr. Zobel:

I'm pleased to inform you that your manuscript has been deemed suitable for publication in PLOS ONE. Congratulations! Your manuscript is now with our production department. 

Kind regards, 

on behalf of

Dr. Sathishkumar V E 

Academic Editor

PLOS ONE